# New Look of EBV LMP1 Signaling Landscape

**DOI:** 10.3390/cancers13215451

**Published:** 2021-10-29

**Authors:** Ling Wang, Shunbin Ning

**Affiliations:** 1Department of Internal Medicine, Quillen College of Medicine, East Tennessee State University, Johnson City, TN 37614, USA; 2Center of Excellence for Inflammation, Infectious Diseases and Immunity, Quillen College of Medicine, East Tennessee State University, Johnson City, TN 37614, USA

**Keywords:** EBV, LMP1, LIMD1, LUBAC, p62

## Abstract

**Simple Summary:**

Epstein-Barr Virus (EBV) infection is associated with various lymphomas and carcinomas as well as other diseases in humans. The transmembrane protein LMP1 plays versatile roles in EBV life cycle and pathogenesis, by perturbing, reprograming, and regulating a large range of host cellular mechanisms and functions, which have been increasingly disclosed but not fully understood so far. We summarize recent research progress on LMP1 signaling, including the novel components LIMD1, p62, and LUBAC in LMP1 signalosome and LMP1 novel functions, such as its induction of p62-mediated selective autophagy, regulation of metabolism, induction of extracellular vehicles, and activation of NRF2-mediated antioxidative defense. A comprehensive understanding of LMP1 signal transduction and functions may allow us to leverage these LMP1-regulated cellular mechanisms for clinical purposes.

**Abstract:**

The Epstein–Barr Virus (EBV) principal oncoprotein Latent Membrane Protein 1 (LMP1) is a member of the Tumor Necrosis Factor Receptor (TNFR) superfamily with constitutive activity. LMP1 shares many features with Pathogen Recognition Receptors (PRRs), including the use of TRAFs, adaptors, and kinase cascades, for signal transduction leading to the activation of NFκB, AP1, and Akt, as well as a subset of IRFs and likely the master antioxidative transcription factor NRF2, which we have gradually added to the list. In recent years, we have discovered the Linear UBiquitin Assembly Complex (LUBAC), the adaptor protein LIMD1, and the ubiquitin sensor and signaling hub p62, as novel components of LMP1 signalosome. Functionally, LMP1 is a pleiotropic factor that reprograms, balances, and perturbs a large spectrum of cellular mechanisms, including the ubiquitin machinery, metabolism, epigenetics, DNA damage response, extracellular vehicles, immune defenses, and telomere elongation, to promote oncogenic transformation, cell proliferation and survival, anchorage-independent cell growth, angiogenesis, and metastasis and invasion, as well as the development of the tumor microenvironment. We have recently shown that LMP1 induces p62-mediated selective autophagy in EBV latency, at least by contributing to the induction of p62 expression, and Reactive Oxygen Species (ROS) production. We have also been collecting evidence supporting the hypothesis that LMP1 activates the Keap1-NRF2 pathway, which serves as the key antioxidative defense mechanism. Last but not least, our preliminary data shows that LMP1 is associated with the deregulation of cGAS-STING DNA sensing pathway in EBV latency. A comprehensive understanding of the LMP1 signaling landscape is essential for identifying potential targets for the development of novel strategies towards targeted therapeutic applications.

## 1. Introduction

Human herpesviruses are of particular importance in medical research in that they are associated with severe diseases including cancers in immunocompromised populations [1,2]. Epstein–Barr Virus (EBV), known as human herpesvirus 4 (HHV4), serves as a fascinating paradigm for the study of herpesvirus infection, replication, latency, and associated diseases, as well as host–pathogen interactions due to its ability to establish lifelong persistent infection in normal immunocompetent healthy hosts [3]. EBV establishes lifetime persistence in both IgD^+^CD27^+^ non-class switched and IgD^-^CD27^+^ class switched memory B cells, but not in naïve B cells [4]. EBV entry receptors include CD21, CR1, and CR2, and MHCII in B cells, and αvβ integrins and the newly identified Ephrin receptor A2 in epithelial cells [5,6].

As the first identified human cancer virus, EBV infection is associated with various lymphomas and carcinomas, which most often occur in immunocompromised patients [7]. EBV replication has two modes: one is lytic replication that produces virions and lyses the host cell, and the other is proliferative replication that occurs when the infected B cell proliferates. EBV lytic replication seems to be the cause of Hodgkin lymphomas (HL) [3]. Its latent infection in B cells results in the development of endemic Burkitt’s Lymphoma (BL), Post-Transplant Lymphoproliferative Disorder (PTLD), and Non-Hodgkin Lymphomas (NHL), and in epithelial cells results in Nasopharyngeal Carcinoma (NPC) and some forms of gastric carcinoma [8,9]. In its latency, EBV resides in the infected cell in either integrated or episomal form, with three distinct processes: viral persistence, controlled viral latent gene expression, and the potential to be reactivated from latency [10]. EBV, the other gammaherpesvirus Kaposi’s Sarcoma-Associated Herpesvirus (KSHV/HHV8), and Human papillomavirus (HPV), are three oncogenic viruses causally involved in HIV/AIDS-associated malignancies [11,12]. More than 50% of AIDS-related lymphomas (ARLs), which are a leading cause of HIV/AIDS-related cancer deaths even in the era of combined antiretroviral therapy [13], are associated with EBV infection [14,15].

EBV infection is also associated with autoimmune diseases, including Infectious Mononucleosis (IM), Multiple Sclerosis (MS), Rheumatoid Arthritis (RA), and Systemic Lupus Erythematosus (SLE) [16,17], underscored by the fact that EBNA1 was initially identified as a seropositive EBV antigen in RA patients [18]. More recently, EBV has been detected in astrocytes and microglia in the brain of more than 90% (91/101) of MS patients versus 24% (5/21) of non-MS neuropathologic patients [19], which may represent a new mechanism that underlies the association of EBV with Mild Cognitive Impairment (MCI) or Alzheimer’s Disease (AD), supporting the hypothesis that EBV infection promotes cognitive dysfunction [20].

The EBV Latent membrane protein 1 (LMP1) is a pleiotropic factor, which is constitutively active without the need of ligand stimulation and plays diverse roles through the whole EBV life cycle, including lytic, reactivation, and latency. As the principal oncoprotein of EBV, LMP1 is essential for transformation and proliferation in multiple cell backgrounds in vitro as well as in transgenic mice, through regulating a vast scale of cellular functions such as proliferation and survival, apoptotic resistance, immune modulation, anchorage-independent growth, metabolism, angiogenesis, metastasis, and invasion, in a context-dependent manner. Other EBV products, however, play complementary roles in regulating these processes to promote oncogenesis [21,22].

In this review, we summarize recent research progress on LMP1 signaling, focusing on the novel components of LMP1 signalosome, including LIMD1, p62, and LUBAC, and the novel functions of LMP1 signaling, including induction of p62-mediated selective autophagy, regulation of metabolism, induction of extracellular vehicles, and activation of Keap1-NRF2-mediated antioxidative defense.

## 2. LMP1 Structure and Novel Components of LMP1 Signalosome

The LMP1 protein N-terminus has six Transmembrane Domains (TMs) that are majorly responsible for LMP1 membrane tethering and oligomerization. The long C-terminus in the cytoplasm can be functionally divided into two well-known domains, C-Terminal Activation Region 1 (CTAR1) and CTAR2, and a third less-understood domain CTAR3.

Decades have been spent on the study of LMP1 signaling pathway, with numerous essential intermediators, such as TRAFs and MAPKs, being identified for the activation of NFκB, AP1, and Akt. LMP1 CTAR1 directly interacts with TRAF1, -2, -3 and -5, and indirectly interacts with TRAF6 via TRAF2 and -5; CTAR2 directly interacts with TRAF2, and indirectly interacts with TRAF6 that is in a complex containing TRAF2, TRADD, RIP1, and other potentially unidentified components [23,24,25].

As a member of the tumor necrosis factor receptor (TNFR) superfamily, LMP1 shares many signaling components with TNFR and CD40, as well as with pathogen recognition receptors (PRRs), and both LMP1 and PRRs activate a subset of Interferon Regulatory Factors (IRFs), which, along with NFκB, AP1, and Akt, all contribute to EBV-mediated pathogenesis [24,25,26,27,28,29]. LMP1 also shares a spectrum of signaling components with Human T-cell Leukemia Virus-1 (HTLV1) Tax for NFκB/AP1 activation [30,31].

Recent efforts on the study of LMP1 signaling include the identification of BS69, an adaptor containing PHD, Bromo, PWWP, and MYND domains, which recruits TRAF6 to LMP1 leading to AP1 activation [32]. Later, BS69 was shown to compete with TRADD for LMP1 binding, and therefore represses LMP1 CTAR2 activation of canonical NFκB [33]; BS69 also represses LMP1 CTAR1 activation of noncanonical NFκB through its direct interaction with TRAF3 [34]. Furthermore, we have identified several novel components of LMP1 signalosome in recent years, including the linear ubiquitin (Ub) assembly complex (LUBAC), the adaptor protein LIMD1, and the Ub sensor and signaling adaptor p62.

### 2.1. The Linear Ubiquitin Assembly Complex LUBAC

Post-Translational Modifications (PTMs) by ubiquitin (Ub) or Ub-like small regulatory proteins are a pervasive topic important to the regulation of activation, stability, and functions of many transcription factors and other proteins in various processes [35,36]. It is clear that targeting of cellular proteins for K48 ubiquitination-mediated proteasomal degradation is an important aspect of infection and cell transformation by tumor viruses including EBV [37]. Further, atypical ubiquitination, represented by K63-linked nonproteolytic polyubiquitination, has become a key mechanism in a myriad of biological functions [38,39,40].

EBV encodes several viral proteins that exploit the host ubiquitin system to regulate its latency and persistence in the host cell [37,41,42]. Both regulatory and degradative ubiquitination forms are involved in LMP1 activation of downstream transcription factors. We have shown that K63-linked regulatory ubiquitination contributes to LMP1 activation of IRF7, with the requirement of RIP1 and TRAF6 [43,44]. Later, we have further identified TNFAIP3 (also known as A20) as an important de-ubiquitinating enzyme (DUB) that negatively regulates IRF7 ubiquitination stimulated by LMP1, and therefore plays an important role in balancing IRF7-mediated functions in EBV latency [45]. These intriguing findings indicate that EBV utilizes both ubiquitination and deubiquitination machinery to subvert the host cellular pathways.

LUBAC is a ternary ubiquitin ligase complex composed of HOIP (RNF31), HOIL1L (RNF54), and SHARPIN (Figure 1), with RNF31 being the central component [46]. LUAC complex is constitutively formed under normal physiological conditions [47,48,49]. LUBAC-mediated linear polyubiquitination has come into focus in the past years due to its emerging role in specific activation of NFκB, but not JNK, by conjugating linear polyubiquitin chains onto NEMO and RIP1 [50,51], in response to diverse signaling stimuli [50,52,53,54,55,56], including apoptotic and immune stimuli such as TNFα [47,49], IL1β [57], genotoxic stress [58], CD40 [59], Toll-Like Receptors (TLRs) [60], NOD2 [61], and NLRP3 [62]. However, LUBAC negatively regulates RIG-I-mediated innate immune responses by targeting RIG-I, TRIM25, and IRF3 for degradation [63,64], and by disrupting the TRAF3-MAVS complex [65].

The involvement of LUBAC in EBV infection is also emerging [66,67]. A high throughput screen identified the association of LUBAC with TRAF1, which is induced by LMP1 in EBV latency [66]. Further investigation has revealed that LMP1 CTAR1 induces K63-linked ubiquitination of TRAF1 in the TRAF1:TRAF2 complex, which further facilitates the recruitment of LUBAC to LMP1 [67].

Importantly, we have shown that the key LUBAC component RNF31 interacts with LMP1 and IRF7 in EBV-transformed cells. Consequently, LUBAC stimulates linear ubiquitination of NEMO and IRF7, promoting NFκB activity but inhibiting IRF7 activity downstream of LMP1 signaling [68]. These findings implicate the importance of LUBAC in controlling LMP1-mediated pathways for its pathogenic functions and may have broad significance on IRF7-mediated IFN response in antiviral immunity. Likewise, a later study has shown that HTLV1 Tax also invokes LUBAC-mediated linear ubiquitination for NFκB activation [69].

### 2.2. The Adaptor Protein LIMD1

LIMD1 is a member of the ZYXIN gene family that includes AJUBA, TRIP6, LPP, WTIP, migfilin, and ZYXIN. As an adapter, LIMD1 possesses multiple protein-interacting domains (Figure 2A), which are able to interact with various proteins, including Rb [70], TRAF6 [71], p62/SQSTM1 [72], VHL and PHD [73], to regulate different cellular processes [72]. For example, the interaction of LIMD1 with TRAF6 enhances the ability of TRAF6 to activate AP1 and negatively regulates the canonical Wnt receptor signaling pathway in osteoblasts [71]. LIMD1 also positively regulates microRNA (miRNA)-mediated gene silencing by interacting with the microRNA-Induced Silencing Complex (miRISC) [74].

Interestingly, our GEO high throughput profiling has revealed that LIMD1 is associated with IRF4 at the transcriptional level in B-cell lymphomas, including EBV-associated lymphomas [75]. Further analyses have demonstrated that IRF4 and NFκB, both oncogenic transcription factors activated by EBV LMP1 and HTLV1 Tax signaling pathways, transcriptionally upregulate LIMD1 expression in EBV- or HTLV1-transformed cells [76]. More importantly, LIMD1, in turn, interacts with TRAF6 and participates in LMP1 signal transduction for the activation of downstream NFκB and AP1. LIMD1 depletion impairs LMP1 signaling and functions, potentiates ionomycin-induced DNA damage and apoptosis, and inhibits p62-mediated selective autophagy that we have later demonstrated to be induced by oxidative stress in oncoviral latency [76,77]. Thus, LIMD1, known as a tumor suppressor in several tumors such as lung, gastric, and breast cancers [78,79,80], oppositely acts as a tumor promoter in viral hematomalignancies.

### 2.3. The Ubiquitin Sensor and Signaling Hub p62

p62 (also named EBIAP, ZIP3, SQSTM1/Sequestosome-1), a human homolog of mouse Zeta PKC-interacting proteins (ZIPs), functions as a signaling hub that comprises multiple protein-interacting domains and is able to interact with different proteins in distinct signaling pathways including LIMD1, TRAF6, LC3b, Keap1, etc. (Figure 2B), to control myriad cellular processes, including osteoclastogenesis, obesity, cancer development, DNA damage response (DDR), aging, inflammation and immunity, autophagy, and oxidative stress [81,82]. It is known that p62 is upregulated in several types of cancer, such as breast, lung, and liver tumors [83,84].

Of note, p62 C-terminus is a ubiquitin-binding region (UBA) that enables its function as a “ubiquitin sensor” (Figure 2B). The UBA binds to K63 ubiquitin chains to facilitate NFκB activation in diverse contexts [81,85]. Like LIMD1, as a signaling hub protein, p62 also has a TRAF6-binding domain (Figure 2B), which specifically recognizes TRAF6, but not TRAF5 or TRAF2. Binding of p62 to TRAF6 facilitates TRAF6 K63-linked polyubiquitination, in which the N-terminal dimerization domain and the UBA domain of p62 are also required [86,87,88]. Thus, p62 has at least two roles in activating NFκB. First, p62 probably favors the phosphorylation of IKKβ by functioning as a ubiquitin receptor that facilitates the recruitment of ubiquitinated signal intermediators. Second, p62 facilitates TRAF6 K63-linked polyubiquitination through interacting with TRAF6. In addition, the PB1 domain of p62 interacts with MEKK3 to regulate NFκB activity [89].

Our most recent study has identified p62 as another novel component of LMP1 signalosome, like LIMD1, also through its interaction with TRAF6, with LIMD1 binding to the N-terminal RING domain and p62 to the C-terminal TRAF domains (Figure 2C). Consequently, p62-TRAF6 interaction potentiates TRAF6 ubiquitination, further promoting activation of downstream NFκB, AP1, and Akt [90].

## 3. New Insights into LMP1-Mediated Pathogenesis

LMP1 is well known to activate the transcription factors NFκB and AP1, as well as HIF1α, T-Cell Factor (TCF) and STATs. We have expanded this list by adding IRF7 [43,44] and IRF4 [91,92]. Moreover, we have collected evidence suggesting that the master antioxidant transcription factor NRF2 can also be activated by LMP1 pathway (to be published). LMP1 exerts most of its known functions via these transcription factors, which transcriptionally regulate the expression of unique targets in distinct biological processes, with some functions being context dependent.

Besides the means of activation of transcription factors, LMP1 also takes many other strategies to perturb, reprogram, and regulate a large range of host cellular mechanisms and functions, which have been increasingly disclosed.

### 3.1. LMP1 Induces Both Random and Selective Autophagy Programs

Autophagy, with either non-selective or selective form, is one of the two major intracellular protein degradation mechanisms, which are essential for proteostasis in eukaryotes [93,94]. Although debates currently exist on the segregation of these two forms, selective autophagy, such as mitophagy, is mediated by an increasing pool of receptors, including p62 [82,95,96]. The two major degradation mechanisms, autophagy and Ub-Proteasome System (UPS), have reciprocal crosstalks; however, they are different in many ways. UPS targets K48-linked ubiquitinated, short-life, soluble misfolded proteins to proteasomes for degradation, whereas autophagy targets non-ubiquitinated (random autophagy) or ubiquitinated (selective autophagy), long-life, insoluble misfolded proteins or whole organelles to lysosomes for degradation [97,98].

Autophagy plays a dual role in cancers; as either tumor suppressor at early stage or promotor at late stage [99,100,101,102,103,104]. Oncogenic viruses, including EBV, are known to inhibit autophagy, which serves as an immune defense strategy [105,106], at their early stage of infection for optimal replication and oncogenic transformation [107,108,109,110,111], but induce autophagy in their latency to facilitate their persistence and tumorigenesis through different mechanisms, including regulation of metabolism and DDR [112,113,114].

High physiologic levels of LMP1 is able to induce the Unfolded Protein Response (UPR)/ Endoplasmic Reticulum (ER) stress response via the N-terminal six TMs by activating all three branches of the UPR (PERK, ATF6, and IRE-1) in a sequential manner [115,116,117], which further induce autophagy, and in turn, high levels of autophagy promote LMP1 degradation [118,119]. Moreover, higher levels of LMP1 induce reactive oxygen/nitrogen species (ROS/RNS, referred to ROS hereafter), which trigger DDR that can also induce autophagy [120]. LMP1 also induces the expression of autophagy-related genes such as Bcl2A1, and might induce the formation of autophagosomes via induction of PI3K/mTOR that is activated by LMP1 CTAR1 [121]. Interestingly, we have collected preliminary data suggesting that LIMD1, which is induced by LMP1 [76], is also involved in autophagosome formation in response to ROS in EBV latency. In agreement with this, our recent study shows that LIMD1 is associated with a panel of proteins involved in membrane trafficking at the transcriptional level in non-small-cell lung carcinoma [78].

Furthermore, our recent study shows that LMP1 induces p62 expression in EBV latency, licensing the induction of p62-mediated selective autophagy (Figure 3) [77]. As such, endogenous p62(S403) phosphorylation, p62-LC3b interaction, and p62-autophagosome colocalization, which are unique requirements for the induction of p62-mediated selective autophagy in addition to p62 transcriptional upregulation and p62(K420) ubiquitination [122,123,124,125], are all readily detectable in EBV type 3 latency in association with endogenous ROS levels [77]. These findings indicate that EBV latent infection induces p62-mediated selective autophagy via ROS-mediated mechanism, in which LMP1 at least contributes to p62 expression and ROS production. LMP1 may be also required for PTM-mediated p62 activation (i.e., phosphorylation and ubiquitination), which is under our further investigation.

Functionally, our results show that p62-mediated selective autophagy protects virus-transformed cells from oxidative stress-induced DNA damage at least by confining p62 in the cytoplasmic compartment. Inhibition of autophagy causes p62 translocation to and accumulation in the nucleus, where p62 inhibits DNA damage repair through its ability to target DNA repair proteins CHK1 and RAD51 for proteasomal degradation and its interaction with RNF168 [77], a key Ub E3 ligase for chromatin ubiquitination for activation of both Homologous Recombination (HR) and Non-Homologous End Joining (NHEJ) mechanisms for repairing Double-Strand Breaks (DSBs) [126]. The selective targets other than p62 for p62-mediated selective autophagy in EBV latency remain to be identified.

### 3.2. LMP1 Reprograms Multiple Metabolism Pathways

Cancer metabolism is an old theme, but it has attracted intensive attention in recent years with deeper mechanistic insights. Not only ER-mediated UPR, which can be induced by high levels of LMP1, is involved in regulating lipid and glucose metabolism [127,128], but also LMP1 regulates mitochondrial NAD (NAD+ and NADH) and NADP (NADP+ and NADPH) energy metabolism as well as other metabolism pathways [129,130,131].

Mitochondrial oxidative phosphorylation (OxPhos) and cytoplasmic glycolysis are the two major pathways for energy production in a normal cell, but cancer cells usually take the latter strategy to produce a large amount of lactate even in the presence of oxygen, which is termed aerobic glycolysis (Warburg effect), a cancer hallmark characterized by increased glucose consumption and decreased OxPhos. LMP1 promotes the Warburg effect by directly potentiating the expression, stability, and plasma localization of GLUT1, and consequently induces the expansion of Myeloid Derived Suppressor Cells (MDSCs) in NPC [132,133]. LMP1 also promotes glycolysis by constitutively activating the FGF2-FGFR1 signaling pathway, and by upregulating the glycolysis-related hexokinase 2 (HK2) and IDH2, and HIF1α-mediated PDK1 and PKM2, and by repressing the expression of HOX genes [131]. Furthermore, in EBV lymphomas, LMP1 promotes glucose metabolism through stabilization of c-Myc, promoting c-Myc activity, and also through upregulation of c-Myc expression via STAT3 [129].

LMP1 activates PI3K, which further activates Akt and mTOR1, the master regulators of glucose metabolism. In fact, the PI3K subunit, PIK3CA, is hyperactivated by somatic mutation in epithelial cancers, including >80% EBV-positive gastric carcinomas.

More recently, it has been revealed that metabolic rewiring is beyond the Warburg effect in many cancer cells. Mitochondrial NAD/NADP metabolism, nucleotide and fatty acid synthesis, glutaminolysis, and mitochondrial one carbon metabolism pathways are also deregulated in cancer cells to support proliferation, malignant outgrowth, and survival [130,131]. Myc, upregulated by EBNA2 and LMP1 in EBV lymphomas, promotes one carbon metabolism that contributes to nucleotide synthesis, NADPH production, and antioxidative defense [134]. LMP1 also regulates NAD/NADP metabolism by upregulating the NAD(P)H oxidase NOX [135].

### 3.3. LMP1 Regulates Epigenetics by Promoting Histone and DNA Methylation

Apart from regulating individual genes via phosphorylation-mediated activation of a panel of transcription factors, LMP1 is also able to regulate global gene expression by promoting chromatin and DNA methylation.

PARP1 is a crucial player in DNA damage repair, NAD metabolism, and chromosome 3D architecture conformation. LMP1 promotes PARP1 activation that requires high levels of NAD+, likely mediated by ERK, in B lymphocytes and epithelial cells [136]. Upon activation, PARP1 promotes histone PARylation to alter chromosome 3D architecture in cooperation with CTCF, which is important for maintenance of EBV latency [136,137]. PARP1 also promotes replication stress at oriP, in a manner depending on the cell cycle [138].

Global CpG DNA hypermethylation (CIMP) is a unique feature of epithelial cancers. LMP1 transcriptionally upregulates the DNA methyltransferase I (DNMT1) and potentiates its activity [139], and in turn, DNMT1 and its partner UHRF1 restricts the expression of LMPs and EBNAs through CIMP [140], in addition to a pool of cellular tumor-suppressor genes as week as B-cell activation antigens that are also downregulated through CIMP [141]. LMP1 also regulates DNMT1 mitochondrial localization, which consequently silences pTEN gene expression and downregulates the OxPhos complexes by promoting hypermethylation of mitochondrial DNA (mtDNA), leading to metabolic reprogramming in NPC [139].

The universal methyl donor S-adenosylmethionine (SAMe) for DNA, protein, and lipid methylation is produced by metabolism from ATP, serine, methionine, and vitamins B9 and B12. Thus, high metabolic rates, which are partially contributed by LMP1, are required for epigenetic regulation in EBV latency. EBV induces high levels of MTHFD2 and SHMT2, two key enzymes in methionine and serine metabolism, respectively, during B-cell immortalization [134].

### 3.4. LMP1 Promotes Formation of Extracellular Vehicles

LMP1 has been long recognized as a membrane protein tethering to the lipid rafts with its N-terminal TM domains, of which the region 38-FWLY-41 in TM1/2 is critical for membrane tethering, LMP1 dimerization, and signaling [142]. In EBV-transformed B lymphocytes, a small portion of LMP1 protein undergoes phosphorylation, which is preferentially associated with vimentin in the cytoskeleton network likely mediated by TRAF3 that interacts with CTAR1 [143,144].

Opposite to the conventional doctrine that LMP1 signals from the plasm membrane, later studies have shown that LMP1 principally signals from intracellular compartments containing lipid rafts in different cell types [145]. Tethering LMP1 to the lipid rafts and the vimentin cytoskeleton facilitates PI3K localization and consequent signal transduction to Akt and ERK activation [146]. LMP1 also undergoes palmitoylation at cysteine 78, but palmitoylation is not required for its localization to the membrane [143,147].

Besides intracellular compartments, interestingly, LMP1 is also enriched in exosomes isolated from EBV-infected B cells and epithelial cells, and can be transferred to uninfected cells by these extracellular vesicles [148]. Moreover, LMP1 stimulates biogenesis and secretion of extracellular vesicles, with the requirement of CD63 [121,149]. In this regard, LMP1 utilizes extracellular vesicles to promote cell growth, invasion, and metastasis. Extracellular vesicles are also believed to associate with the immune modulation function of LMP1, underscored by the fact that they play a role in antigen transfer [148,149,150,151,152].

### 3.5. LMP1 Regulates Antiviral and Antitumor Immune Responses

Accumulating evidence has shown that LMP1 regulates both innate and adaptive immune responses at discrete stages and multiple layers [16,153,154,155]. As stated above, LMP1 induces autophagy that, as an intrinsic immune defense mechanism, intimately crosstalks with the immune system, for instance, enhancing antigen presentation to expose the infected cells to the immune system [156,157,158], and inducing the formation of exosomes that plays a role in antigen transfer. Expression of LMP1 in mouse B cells enhances antigen presentation and costimulation through CD70 and OX40L [159], consequently driving potent cytotoxic CD4+ and CD8+ T cell responses [160]. Moreover, LMP1 plays a crucial role in the regulation of apoptosis, which also serves as a defense strategy in antiviral and antitumor responses [161].

LMP1, like PRRs, activates NFκB, AP1, and Akt by promoting their site-specific phosphorylation, and we have added IRF7 and IRF4 to this list of its activated targets (Figure 3) [43,44,92], all of which participate in the regulation of innate immune response [162,163]. As to IRF4, it is known to inhibit type I IFN-mediated response, and inhibits IRF5 expression and activity in the context of EBV infection [164,165]. We and others have also shown that LMP1 induces the expression of IRF7 and IRF4 via NFκB in EBV-infected cell lines as well as in LMP1 transgenic mice [159,165,166,167]. Moreover, LMP1 CTAR3 stabilizes IRF7 and limits its transcriptional activity by promoting its sumoylation (Figure 3) [168]. We have further shown that IRF5, the dominant-negative mutant of which is induced by TLR7 signaling in EBV latency, interacts with IRF7 and inhibits its activity stimulated by LMP1 [169]. In addition, LMP1 is a potent inhibitor of TLR9 transcription [170], and promotes RIG-I proteasomal degradation [171].

The type I IFN Jak-STAT pathway, which is responsible for the robust production of type I IFNs, is substantially compromised if not completely disabled in EBV latency, while the complex mechanisms are poorly understood. We have conducted high throughput phosphoproteomics to profile the deregulated phosphorylation in Jak-STAT pathways, and the results have shown that STAT2 Y690 phosphorylation was most downregulated and barely detected in EBV latency. IFNα treatment or virus infection restored STAT2 Y690 phosphorylation in cell lines lacking LMP1 but did not or only weakly restored in cell lines expressing high levels of LMP1, strongly suggesting a role for LMP1 in repressing the type I IFN Jak-STAT pathway. Consistent with these preliminary results, it has been reported that LMP1 N-terminal TM interacts with Tyk2, and consequently, suppresses phosphorylation of both STAT1 and -2 and subsequently blocks type I IFN-mediated antiviral responses [172]. Our phosphoproteomics results imply that all other Jak-STAT pathways, however, are active in EBV latency [173].

In addition, we have preliminary data in an ongoing project showing that LMP1 plays a role in the regulation of the cGAS-STING DNA sensing pathway.

In summary, LMP1 regulates both innate and adaptive immunity in both positive and negative manners, thus playing multiple roles in the fine balance of the complicated host–virus interaction for EBV long-term persistence and pathogenesis. Further investigation to disclose important but unknown strategies that LMP1 employs to deactivate type I IFN Jak-STAT pathway is one of our current plans.

### 3.6. LMP1 Promotes Chronic Inflammation and the Tumor Microenvironment

It is also known that LMP1 functions as an inflammation-promoting factor, at least by inducing a panel of pro-inflammatory cytokines, such as TNFα, IL6, IP10, among many others, via NFκB, AP1, and STAT3 [174,175,176]. Activation of NFκB and AP1 by LMP1 also induces miRNAs including miR-155, miR-146a, and miR-21, which play key roles in the regulation of inflammation [177,178,179]. We have shown that LMP1 induction and activation of IRF4 contributes to BIC/miR-155 expression in EBV latency [92,180].

Several novel mechanisms accounting for LMP1-mediated chronic inflammation are emerging, including its ability to balance oxidative stress and NRF2-mediated antioxidant defense, to balance DNA damage and DNA repair at least via induction of autophagy, to transfer to extracellular compartments via extracellular vehicles, and to reprogram extra-mitochondrial glycolysis metabolism by promoting the expression of multiple glycolytic genes (e.g., GLUT1), which further promotes MDSC expansion to facilitate immune escape in the Tumor Microenvironment (TME) and promotes the expression of NLRP3 inflammasome [133]. All these events contribute to the development of chronic inflammation, as discussed elsewhere in this review.

As such, the indispensable roles of LMP1 in regulating redox homeostasis, chronic inflammation, and immune response render its ability to contribute to the development of the TME, as discussed elsewhere in the review.

### 3.7. LMP1 Inhibits DNA Damage Response and Promotes Genomic Instability

DNA damage is directly linked to many human diseases including cancer. Thus, eukaryotic organisms have developed sophisticated mechanisms to repair damaged DNA to secure genomic integrity. HR and NHEJ are two major mechanisms responsible for repairing DSBs [181]. Most cancers, if not all, harbor deficient traditional DNA repair mechanisms [182].

Viruses can manipulate host DDR machinery in that infected cells recognize viral replication as DNA damage stress [183,184,185,186]. In the absence of active viral replication, ROS produced in viral persistence are the major cause of different levels of endogenous DNA lesions [187]. It is well documented that LMP1 inhibits DNA repair to promote genome instability and renders DNA damage resistance in different cell contexts [188,189,190,191]. So does HTLV1 Tax [30,192,193]. However, the underlying mechanisms are not fully understood. 

As one of these mechanisms, p62-mediated selective autophagy plays an alternative and indispensable role in DNA repair in cancer cells [126,194,195,196]. Importantly, our recent discovery of the induction of p62-mediated selective autophagy by LMP1-ROS in EBV latency has defined two distinct roles for the autophagy-p62 interplay in DDR in this setting: cytoplasmic p62, which is partially induced by LMP1 [90], mediates selective autophagy (in addition to mediating LMP1 signal transduction [90]), and nuclear p62 accumulated upon autophagy inhibition represses DNA repair. These original findings indicate that a well-controlled autophagy-p62 interplay renders EBV-positive cells with the ability to balance pro-survival DNA damage resistance and pro-mutagenic genomic perturbation under oxidative stress [76,77].

### 3.8. LMP1 Facilitates Telomere Integrity and Represses Premature Cell Senescence

One of the mechanisms accounting for EBV oncogenesis is the ability of the virus to repress premature cell senescence in its latency, which involves LMP1 regulation of telomere/telomerase and cell cycle regulators such as p16INK4a, p21CIP1/WAF1, and CDKN2A/B [21,197,198,199].

Telomere integrity is a prerequisite for the cell to escape senescence during EBV immortalization [21]. LMP1 stable overexpression in EBV-negative BL cells causes telomere de-protection and dysfunction, as well as multinucleation, by negatively modulating telomerase-independent telomere elongation named alternative lengthening of telomere (ALT) through downregulation of shelterin components including the telomere repeat binding factors TRF1, TRF2 and POT1, which were also found in newly EBV-transformed B cells involving distinct mechanisms mediated by several viral proteins other than LMP1, including BNRF1 and EBNA1 [200,201]. Consequently, these deregulations may favor chromosomal instability, cell immortalization, and oncogenic transformation. However, in LCLs, LMP1 promotes telomere elongation and protection through upregulation of telomerase activity that involves distinct mechanisms at both the transcriptional and translational levels [21,199]. Of note, LMP1 transcriptionally activates the TERT gene to induce telomerase expression, which involves NFκB, MAPK, and ERK pathways, but the involvement of c-Myc is controversial that may be cell type-dependent [202,203].

Consistent with its ability to promote malignant transformation and immortalization, LMP1 represses mouse fibroblast replicative senescence in vitro by promoting CRM1-dependent nuclear export of Ets2 that is a transcription factor for p16INK4a, consequently inhibiting p16INK4a expression [204,205], in which LMP1 CTAR2 TRADD-binding domain is required [206], and also by blocking downstream mediators of the p16INK4a-Rb pathway [205]. Counterintuitively, chronic NFκB activity induced by HTLV1 Tax was reported to promote p21/p27-dependent senescence response, which is overcome at least by the viral antisense protein HBZ [207,208].

### 3.9. Potential Role of LMP1 in Activation of the Master Antioxidant Defense

The discovery of the cellular mechanism of oxygen sensing has been awarded the 2019 Nobel Prize in Physiology and Medicine, highlighting the importance of oxygen homeostasis in cellular functions. ROS at low or moderate levels act as important second messengers for normal cellular functions but are harmful at high levels. Thus, ROS levels are strictly balanced by several detoxification processes mediated by antioxidant enzymes to ensure normal cellular functions, and this balance is called “redox homeostasis”. Loss of this balance is called “oxidative stress” [209,210].

ROS at abnormally high levels are the major cause of endogenous DNA damage [211,212,213], which can incite inflammation, and excess inflammation, in turn, causes oxidative stress, ultimately resulting in tissue damage [211]. DNA damage also perturbs genomic instability that promotes malignant transformation under certain conditions [187,214]. Abnormal elevation of ROS levels and chronic inflammation are the most common features of persistent (chronic and latent) viral infections [215,216]. Aberrant redox homeostasis is one of the hallmarks of cancer [217,218]. ROS represent a crucial component of tumor niches [219,220], and activate various transcription factors such as NFκB, AP1, HIF1α, and STAT3 that are essential for cancer initiation and development. ROS also control the expression of various tumor suppressor genes such as p53, Rb, and pTEN [221]. Moreover, ROS promote cancer development by inducing autophagy [222]. Elevated ROS levels exist in almost all cancers, if not all, as well as in EBV latency [223]. It is well known that LMP1, among several other EBV products, contributes to the majority of ROS production in EBV latency, at least via JNK-p38/AP1 [223,224]. Mitochondria produce the majority of ROS [225]. The mitochondria in malignant cells are functionally and structurally deregulated and are able to overproduce ROS, thus playing a central role in ROS-caused DNA damage [226].

The transcription factor NRF2, a member of the basic leucine zipper (bZIP) family, is the master regulator of oxidative and inflammatory stresses by transactivating about 250 genes, of which many are involved in antioxidant defense including p62, Keap1, Cox2, iNOS, PRDX1, HIF1, NQO1, HMOX1, GSTs, etc., as well as NRF2 itself [227,228]. NRF2 target genes are also involved in the regulation of immune responses and inflammation, metabolism, cell proliferation, survival and anti-apoptosis, angiogenesis, cell migration and invasion, all of which contribute to cancer development [229,230,231,232]. Under normoxia, NRF2 is constitutively synthesized but quickly ubiquitinated for proteasome-mediated degradation by the Ub E3 ligase complex Keap1/Cul3/RBX1. ROS trigger autophagic degradation of Keap1, consequently resulting in the accumulation and activation of NRF2 (phosphorylation at S40) [233]. Due to its importance in redox homeostasis, quick activation of NRF2 is a hallmark in response to ROS under various stresses [234]. As the master antioxidant defense mechanism, the Keap1-NRF2 pathway plays a central role in the regulation of inflammation, and has a close connection to NFκB-mediated inflammatory regulation [228,230].

However, deregulation of the Keap1-NRF2 pathway by EBV infection has never been reported. Of note, our recent findings strongly suggest that the Keap1-NRF2 pathway is activated in association with LMP1 in EBV latency [76,77]. The contributions of LMP1 to its activation at least include the ROS production that triggers autophagic degradation of Keap1, and the induction of p62 expression that positively regulates the Keap1-NRF2 pathway activity in a feedback loop. The details underlying the activation of the Keap1-NRF2 pathway in EBV latency and whether EBV primary infection and reactivation also activate this pathway are among the priorities of our current investigation. This study will add NRF2 as another crucial transcription factor that is activated downstream of LMP1 signaling and will disclose a crucial role for the LMP1-NRF2 pathway in redox homeostasis in EBV latency and pathogenesis.

Since we have shown that cytoplasmic p62 also mediates LMP1 signal transduction, p62 plays multiple roles in virus-mediated oncogenesis in both cytoplasmic and nuclear compartments. It is a challenge to separate its cytoplasmic roles in LMP1 and autophagy pathways in that they share the same functional domains of p62, in particular, the C-terminal UBA domain for ubiquitin-mediated process is required for enabling both roles. Nevertheless, the PB domain may be specific to p62’s role for autophagy induction since p62 without this domain still interacts with LMP1, and we have shown that the TB domain, which interacts with TRAF6, is specific to LMP1 signal transduction [90]. In addition, site-specific phosphorylation and ubiquitination of p62, which are essential for p62 function in autophagy, may not be required for LMP1 signal transduction. These details of mechanism to discriminate p62’s roles are under our investigation.

## 4. Perspectives

Identification of novel components is the key to decipher the complicated LMP1 signaling and functions, which would advance mechanistic insights into EBV infection and pathogenesis. It is our understanding that LMP1 engages different signaling components to connect with and manipulate diverse cellular processes. For example, LMP1 employs p62 to enable its function for manipulating cellular selective autophagy and antioxidative stress, employs LIMD1 to repress the anti-oncogenic Hippo pathway, besides their roles in activation of NFκB, and employs UCHL1 to regulate Wnt/β-Catenin/TCF proliferation pathway [235]. Thus, continuing efforts are still needed to fully profile the components of LMP1 signalosome, which would involve high throughput screening, followed by molecular confirmation and functional validation. Further, it will be of great interest to employ the CryoEM technique to study the structure of LMP1 signalosome. This technique has been recently applied to the capsids of EBV and other herpesviruses [236,237,238,239].

LMP1 itself has poor immunogenicity, which restricts it to produce a substantial humoral or cellular immune response and renders it with the ability to escape from CD8+ T cell targeting in EBV-positive healthy individuals. Strategies directly targeting LMP1 for therapeutic purposes seem difficult, although some have been made with promising progress, including LMP1-targeting vaccines and antibodies as well as small peptides and molecules [175]. Targeting LMP1 signal transduction and functions may serve as a viable strategy that could discover unique targets with potential clinical applications [240]. For example, autophagy-mediated degradation is a promising strategy to target non-enzymatic proteins for therapy [241]. A comprehensive understanding of LMP1 signaling pathway, such as the identification of novel components of LMP1 signalosome, is thus essential towards this end.

Disclosing how LMP1 regulates unique immune cells and molecules represents a promising strategy for immunotherapy. For example, LMP1 modulates the activities of stroma, NK, and T cells in the TME, to promote cancer cell growth, survival, metastasis, angiogenesis, and Epithelial–Mesenchymal Transition (EMT) [175,176]. We have been collecting evidence showing that LMP1 is associated with the deregulation of the cGAS-STING DNA-sensing pathway, which has potent anti-tumor activity.

As such, we have several ongoing projects, aiming to identify novel components of the LMP1 signalosome and its unknown or underappreciated functions. These studies may reveal potential targets for designing clinical strategies to treat EBV-associated cancers and diseases.

## 5. Conclusions

LMP1 signalosome contains a pool of cellular proteins, with many being unidentified, which furnish LMP1 with diverse roles to deregulate a large spectrum of cellular events in different stages of the whole EBV life cycle as well as in different cell contexts. Some components of LMP1 signalosome may compete with each other to differentiate LMP1’s abilities to regulate different cellular mechanisms in the same context. As such, some of the components likely vary depending on EBV infection stages and cell contexts, which result in context-dependent outcomes, making it possible to design specific strategies for clinical applications. 

## Figures and Tables

**Figure 1 cancers-13-05451-f001:**
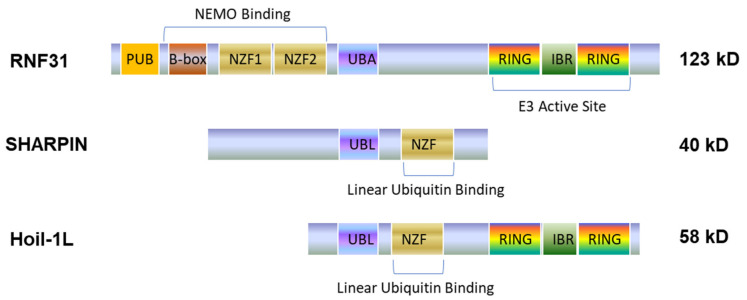
A domain scheme of the LUBAC complex. The LUBAC complex includes three proteins, RNF31, SHARPIN and HOIL1, with RNF31 being the core. PUB: PNGase/UBA or UBX-containing proteins; NZF: Npl4-type zinc finger domain; UBA: ubiquitin-associated domain; IBR: in-between RING domain; UBL: ubiquitin-like domain.

**Figure 2 cancers-13-05451-f002:**
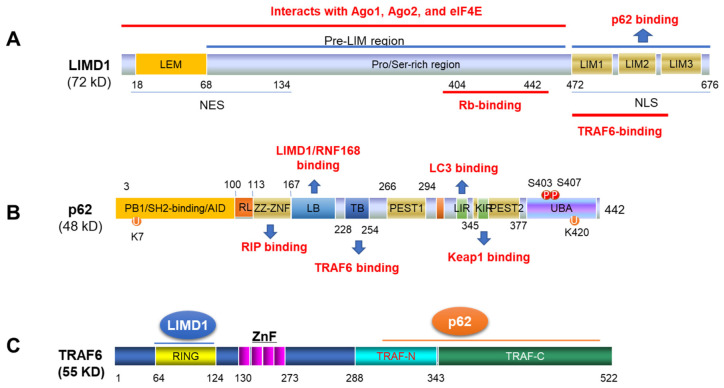
Domain schemes of LIMD1, p62 and TRAF6. The domains of LIMD1 (**A**), p62 (**B**) and TRAF6 (**C**) for their interactions are shown. NES: nuclear export signal; NLS: nuclear localization signal; PB1: Phox/Bem 1p protein–protein binding domain; AID: atypical PKC interacting domain; RL: regulatory linker; ZNF: Zinc finger; LB: LIM protein binding; TB: TRAF6 binding; LIR: LC3-interacting region that mediates interaction with ATG8 family; UBA: Ubiquitin-binding re-gion that binds specifically to K63-linked polyubiquitin chains of polyubiquitinated substrates; NES: nuclear export signal. NLS: nuclear localization signal; ZnF: Zinc finger.

**Figure 3 cancers-13-05451-f003:**
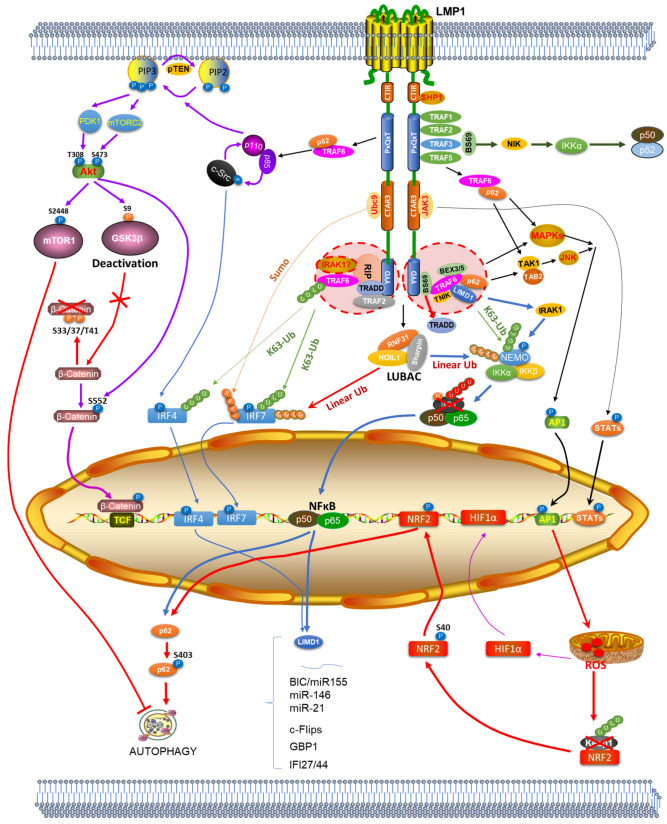
An updated LMP1 signaling pathway. The involvement of LUBAC, LIMD1 and p62 in the LMP1 signal transduction is shown. LIMD1 expression is induced by NFκB and IRF4, and p62 expression is induced by NFκB and NRF2. The induction of p62-mediated selective autophagy and the activation of NRF2-mediated antioxidative defense by LMP1 signaling are shown.

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
