# Peer review of "New Look of EBV LMP1 Signaling Landscape"

_cancers, 2021, doi:10.3390/cancers13215451_

Round 1
Reviewer 1 Report
New Look of EBV LMP1 Signaling Landscape by Wang and Ning is a review article updating the signaling pathways associated with LMP1. The review is detailed and thorough but may be difficult for someone not directly in the field. It add a good amount of new findings to those discussed in earlier reviews.
One suggestion would be to describe the structure of LMP1 in the introduction. A few sentences describing the transmembrane, and CTAR regions would be helpful.
There are some English/grammar areas that need to be addressed.
Page 2. Line 20-21. "Its lytic replication is more likely the cause of Hodgkin lymphomas." This statement should be reworded. As it sands it implies that EBV lytic replication is the causative agent of Hodgkin lymphoma.
Page 2. Line 26. "to reactivation" should probably be "for" reactivation.
Abstract. Line 19-20. "adaptor protein p62, are required" Should probably read "which" are required. Or reworded in some way.
The manuscript should be screened for other minor grammar issue.
Author Response
One suggestion would be to describe the structure of LMP1 in the introduction. A few sentences describing the transmembrane, and CTAR regions would be helpful.
RE: Added as suggested
There are some English/grammar areas that need to be addressed.
Page 2. Line 20-21. "Its lytic replication is more likely the cause of Hodgkin lymphomas." This statement should be reworded. As it sands it implies that EBV lytic replication is the causative agent of Hodgkin lymphoma.
RE: Fixed
Page 2. Line 26. "to reactivation" should probably be "for" reactivation.
RE: Fixed
Abstract. Line 19-20. "adaptor protein p62, are required" Should probably read "which" are required. Or reworded in some way.
RE: Fixed
The manuscript should be screened for other minor grammar issue.
RE: We have kept on the improvement of the writing since submission. We believe the grammar is good enough now
Reviewer 2 Report
The review article by Wang and Ning provides a comprehensive view on the interesting field of LMP1 signaling in the context of Ebstein-Barr virus infection. The authors cover in detail the various aspects of LMP1 signaling. The included figures support the narrative well, but look generic.
Probably, the comprehensiveness of the article is the source of my major concern. This concern is with clarity. The language is impeccably clear and does not need any improvement. However, in the current form the manuscript reads like the unsorted result of a brainstorming meeting. The individual sentences are perfectly understandable and confer a clear message, but neighboring sentences are often not connected by meaning or a clear line of thought. The review presents a lot of facts – as it should – but lacks focus and a clear massage.
A definite outline of the aim of the article, and more importantly a clear, narrowed focus would improve understandability of the manuscript. The authors present 280 references on app. 10 pages of text, leaving not much space to elaborate on the presented findings. According to the subheadings, this concise text aims to cover 12 separate aspects of the LMP1 signaling network.
Both the overstretching scope and the unfocussed narrative leaves the reader more confused than educated.
The authors definitely put in a lot of time and effort to compile every conceivable fact of LMP1 signaling. By now reordering the facts and providing some guidance on how the facts can be interpreted and connected, the authors could make sure that the readers profit from their extensive knowledge about the field.
In detail:
An introductory paragraph should be included that makes it clear to the reader why EBV and LMP1 are important, and what are the essential challenges, controversies and recent findings that the manuscript plans to address. In short: why should the article be read?
The introduction (p1, lines 1-35) lacks a clear line. It is jumpy, and confusing. The reader is alternately - and back and forth - confronted with EBV’s role in multiple sclerosis, various lymphomas, arthritis, etc. This part should be rewritten.
The main text should be thoroughly reordered. This does not only mean moving the subchapters around, but also to reorder the listed facts within these subchapters to yield a clear line of thought. Linking the facts and putting more emphasis on connecting the different aspects addressed in the individual subchapters with suitable lead overs, would also improve understandability.
The authors should consider removing or shorten some aspects they discuss. Maybe less would be more here.
Despite its title the “perspective” does not give an outlook on the direction in which research on LMP1 will or should go. It also is only
The article in itself is a bit too self-referential. The number and placement of self-references is not excessive, but maybe just a bit too much. Terms like “WE have found…” appear very frequently. In addition, the authors’ own preliminary and unpublished data is repeatedly referred to.
Author Response
Probably, the comprehensiveness of the article is the source of my major concern. This concern is with clarity. The language is impeccably clear and does not need any improvement. However, in the current form the manuscript reads like the unsorted result of a brainstorming meeting. The individual sentences are perfectly understandable and confer a clear message, but neighboring sentences are often not connected by meaning or a clear line of thought. The review presents a lot of facts – as it should – but lacks focus and a clear message.
A definite outline of the aim of the article, and more importantly a clear, narrowed focus would improve understandability of the manuscript. The authors present 280 references on app. 10 pages of text, leaving not much space to elaborate on the presented findings. According to the subheadings, this concise text aims to cover 12 separate aspects of the LMP1 signaling network.
Both the overstretching scope and the unfocussed narrative leaves the reader more confused than educated.
The authors definitely put in a lot of time and effort to compile every conceivable fact of LMP1 signaling. By now reordering the facts and providing some guidance on how the facts can be interpreted and connected, the authors could make sure that the readers profit from their extensive knowledge about the field.
RE: We have kept on the improvement of the writing since submission. The manuscript should be better organized and written. We hope these improvements address the reviewer’s major concern.
In detail:
An introductory paragraph should be included that makes it clear to the reader why EBV and LMP1 are important, and what are the essential challenges, controversies and recent findings that the manuscript plans to address. In short: why should the article be read?
RE: Added to the end of “Introduction”
The introduction (p1, lines 1-35) lacks a clear line. It is jumpy, and confusing. The reader is alternately - and back and forth - confronted with EBV’s role in multiple sclerosis, various lymphomas, arthritis, etc. This part should be rewritten.
RE: “Introduction” has been totally re-organized
The main text should be thoroughly reordered. This does not only mean moving the subchapters around, but also to reorder the listed facts within these subchapters to yield a clear line of thought. Linking the facts and putting more emphasis on connecting the different aspects addressed in the individual subchapters with suitable lead overs, would also improve understandability.
The authors should consider removing or shorten some aspects they discuss. Maybe less would be more here.
RE: We have revised the main body substantially. If the reviewer has more thoughts to improve, we would appreciate it in more detail.
Despite its title the “perspective” does not give an outlook on the direction in which research on LMP1 will or should go. It also is only
RE: We have added some specific points in this section.
The article in itself is a bit too self-referential. The number and placement of self-references is not excessive, but maybe just a bit too much. Terms like “WE have found…” appear very frequently. In addition, the authors’ own preliminary and unpublished data is repeatedly referred to.
RE: Yes, we realize this point. But many of the contents of this review are from our group. For example, the whole section 2 are all our findings in recent years.
Round 2
Reviewer 2 Report
The authors made some, but no substantial changes to their manuscript. The article continues to comprehensively cover an interesting - but very specific – field, that might attract a specialized readership.
In the reviewer’s opinion the original manuscript was poorly structured and suffered from overreach by trying to cover every aspect of the limited field in abundant detail. Obviously, the authors do not share these concerns. The revised manuscript has not been re-structured, the scope of the manuscript has not been condensed (although the number of references was reduced from 280 to 240), changes are more or less cosmetically. When changes were actually made: in an apparent attempt to leave the false impression of a strongly reworked manuscript whole stretches of the revised text are marked in red to indicate changes, although the marked parts are identical with the original manuscript.
As the manuscript has not substantially improved – or even significantly changed – the assessment of the manuscript in itself can also not be changed: it is questionable if the potential readership will benefit from such a confusing, unstructured review article.
All concerns with this manuscript are with the quality of its organization, and consequently with its comprehensibility. Scientific concerns – as to be expected with review articles – are not critical. The decision whether to publish the paper is therefore a purely editorial.
As a scientist I have absolutely no problems with authors disagreeing with my assessment of their work and therefore refusing to make changes I suggested. Thus, I would have appreciated if Dr. Wang and Dr. Ning would have clearly stated their disagreement with my opinion and their consequent refusal to change their article. In that case I would have been able to excuse myself from a second review and asked the editor to get an opinion from a new, possibly more positive referee.